# Intensity Switchable and Wide-Angle Mid-Infrared Perfect Absorber with Lithography-Free Phase-Change Film of Ge_2_Sb_2_Te_5_

**DOI:** 10.3390/mi10060374

**Published:** 2019-06-05

**Authors:** Xiaomin Hua, Gaige Zheng

**Affiliations:** 1Jiangsu Key Laboratory for Optoelectronic Detection of Atmosphere and Ocean, School of Physics and Optoelectronic Engineering, Nanjing University of Information Science & Technology, Nanjing 210044, China; Huahuahuaxiaomin@163.com; 2Jiangsu Collaborative Innovation Center on Atmospheric Environment and Equipment Technology (CICAEET), Nanjing University of Information Science & Technology, Nanjing 210044, China

**Keywords:** perfect absorption, phase-change materials, silicon carbide, lithography-free

## Abstract

The range of fundamental phenomena and applications achievable by metamaterials (MMs) can be significantly extended by dynamic control over the optical response. A mid-infrared tunable absorber which consists of lithography-free planar multilayered dielectric stacks and germanium antimony tellurium alloy (Ge_2_Sb_2_Te_5_, GST) thin film was presented and studied. The absorption spectra under amorphous and crystalline phase conditions was evaluated by the transfer matrix method (TMM). It was shown that significant tuning of absorption can be achieved by switching the phase of thin layer of GST between amorphous and crystalline states. The near unity (>90%) absorption can be significant maintained by incidence angles up to 75 under crystalline state for both transverse electric (TE) and transverse magnetic (TM) polarizations. The proposed method enhances the functionality of MMs-based absorbers and has great potential for application to filters, emitters, and sensors.

## 1. Introduction

The realization of a metamaterials absorber (MMA) has long been a goal in engineering the spectral absorption properties of nanostructures [1,2,3]. MMAs have demonstrated the achievement of near-unity absorption in nearly any range of the electromagnetic (EM) spectrum and may be added to the list of unique EM responses achievable with MMs [3,4,5,6]. It is usually difficult to meet the requirements of various specific absorption frequencies, but MMAs can overcome this problem by choosing appropriate structural parameters and integrating active materials [7]. Active control over MMs response can pave the way towards novel optical components like tunable polarizers, sensors, and switches [8,9,10,11,12]. So far, many studies have demonstrated that hybridizations of active media with metallic resonating components can be implemented for further control over the resonant property. Graphene [13,14], liquid crystals [15], and phase-change materials (PCM) [16,17,18,19] have been introduced into MM systems to facilitate a reconfigurable response. The control of graphene permittivity can be achieved with an external gate voltage, thus the integration of graphene into MMs allows for spectrally tunable absorption. Liquid crystals have been used successfully in MMs, as have metasurfaces (MSs) for the tuning of the spectral positions of the resonances using its reorientation by external electric fields. Recently, Tittl et al. used germanium antimony tellurium alloy (GST) to achieve a switchable plasmonic perfect absorber in the mid-infrared ranges (MIR) [20,21].

Spectral controlling and shaping with nanostructures in the MIR region is a hot topic because of the advances in applications such as biochemical sensing, imaging, IR labelling, and so on [22,23,24]. The fast switching between amorphous and crystalline phases of GST makes it a candidate for active MIR photonic devices. The optical properties of GST, specifically its permittivity, vary considerably between amorphous and crystalline phases. Consequently, GST can readily be harnessed and resonantly enhanced in a nanostructure to realize specific switching characteristics [25,26,27]. Moreover, MM- and MS-based absorbers often require rather complex, time-consuming nanofabrication steps, and expensive technical equipment, which limit their practical applications.

In this contribution, we adopted phase-change material (PCM) to design a tunable MIR absorber. The key idea was implemented with an overcladding layer of the PCM GST, which provides the great potential not only to overcome the roadblock of a much easier manufacturing process, but also to dynamically adjust the absorption in the MIR. The switchable and wavelength-selective absorption/emissivity can be accomplished by altering the GST phase states as well as the structural parameters of the Ge/ZnS multilayer. The high contrast in the optical properties of GST enabled thermal tuning of the resonant response with modulation depths of >90%.

## 2. Model and Methods

To approach the construction of an MMA, a numerical investigation into a planar multilayer stack with SiC as the substrate was conducted. As depicted in Figure 1a, PCM GST was deposited on SiC substrate separated by a Ge/ZnS distributed Bragg reflector (DBR). The cross-session view in the *x*-*z* plane is displayed in Figure 1b where *a*, *h*_1_, and *h*_2_ represent the thicknesses of the GST layer, ZnS layer, and Ge layer, respectively. The transverse magnetic (TM) wave is defined as the light source and normally incidents to the stack in this study unless clarified. The GST exhibited two stable structural phases—under the external stimulus, a reversible phase transition between amorphous (aGST) and crystalline (cGST) was experienced. aGST is a lossless dielectric and cGST is a low-lossy dielectric [26]. Within the considered wavelength range (10–12 μm), the refractive index (*n*) of aGST and cGST was approximately 4.3 and 6.3, respectively. As for the extinction coefficient (*k*), aGST was near zero and can be regarded as the transparent dielectric, whereas cGST was about 0.6 and had the relatively higher optical loss [28]. The refractive indices of Ge and ZnS are from Palik’s optical constants handbook [29] and the literature [30], respectively. In the absence of free charge carriers, the optical properties of SiC is given by the Drude–Lorentz model [31]:
(1)$$\varepsilon_{SiC}=\varepsilon_{\infty }\frac{\omega^{2}-\omega_{LO}^{2}+i\gamma \omega }{\omega^{2}-\omega_{TO}^{2}+i\gamma \omega }$$
where $\omega_{LO}$ and $\omega_{TO}$ represent the longitudinal and transverse optical phonon frequencies, chosen as 972 cm^−1^ and 796 cm^−1^, respectively, $\varepsilon_{\infty }$ is set as the high frequency dielectric constant, and γ is defined as the damping rate due to vibrational anharmonicity. In this work, $\varepsilon_{\infty }$ was chosen as 3.75 cm^−1^ and γ was set as 6.5.

The theoretical investigation of this configuration is based on the transfer matrix method (TMM) [32,33]. This method involves two different matrices: One is the transmission matrix and the other is the propagation matrix, which can be described as:
(2)$${\mathrm{TM}}_{l}=\frac{1}{t_{l}}\left[\begin{array}{ll} 1 & r_{l}\\ r_{l} & 1 \end{array}\right],$$
(3)$${\mathrm{PM}}_{l}=\left[\begin{array}{l} \exp\left(-in_{l}d_{l}\omega /c\right)\;0\\ 0\;\exp\left(in_{l}d_{l}\omega /c\right) \end{array}\right].$$

Here, *t_l_* and *r_l_* denote the transmission and reflection coefficients of light transmission from the l-th layer to the (l + 1)-th layer, respectively, which can be derived from the Fresnel equation; *d_l_* is the thickness of the l-th layer. The transmission matrix of this composition can be described as
(4)$$M=\prod_{l}{\mathrm{TM}}_{l}\cdot {\mathrm{PM}}_{l}.$$

Subsequently, the transmission, reflection, and absorption of this composition can be respectively described as T = |1/M_11_|^2^, R = |M_21_/M_11_|^2^, and A = 1 − T − R.

## 3. Results and Discussions

Phase-change switchable MMs typically comprise of a metallic layer structured with an active PCM layer. Either enhancement or suppressant of resonances in the structured layer can be realized. For the GST, the dielectric constants can change dramatically once the phase transition is initiated by the thermal filed. Figure 2 shows the absorption spectrum of the device for both amorphous (blue curve) and crystalline (red curve) GST. During the structural transition, the absorption underwent an intensity enhancement with a peak at the resonance wavelength of 11.34 μm.

Thus, the absorption can switch between two separate phases of GST. The dynamic absorption control in the presented structure represents a distinct advantage over passive infrared absorbers and emitters [22,34,35,36].

The simulated electric field (|E|) distributions on- and off-resonance when the GST is in the amorphous and crystalline phases are shown in Figure 3. It is clear that the strong electric fields were concentrated within a certain region, the field profile of the typical waveguide mode was generated as well. The field enhancement factors at the waveguide-GST interfaces were as high as ~6 under crystalline state (Figure 3b). This increased field intensity led to enhanced absorption at resonant wavelength. Compared with Figure 3a, the low value of the electric field intensity in the Ge/ZnS multilayer with amorphous state resulted in a high reflection.

Absorption spectra of designs with different GST thicknesses (0.15, 0.225, 0.3, 0.375, and 0.45 μm) is shown in Figure 4. The maximum absorption for cGST thickness of 0.15, 0.225, 0.3, 0.375, and 0.45 μm was 0.89, 0.93, 0.94, 0.94, and 0.95, respectively, and the resonance wavelengths were 10.63, 11.09, 11.34, 11.5, and 11.61 μm, respectively. The full width at half-maximum (FWHM) of structures for GST thickness of 0.15, 0.225, 0.3, 0.375, and 0.45 μm were 1.05, 0.76, 0.39, 0.33, and 0.26 μm, respectively, indicating quality factors (Q = λ_0_/Δλ; λ_0_ is the peak wavelength, Δλ is the FWHM) of 10.12, 14.59, 29.08, 34.85, and 44.65, respectively. The numerical results showed that over 90% peak absorption was maintained when *a* ranged from 0.15 to 0.45 μm at the crystalline phases with other parameters fixed. The resonance wavelength met a red-shift with the increase of *a*, thus the spectral selectivity of the structure was improved.

Optical Tamm state (OTS) is a type of surface mode, which can be excited by the normal incidence of light with a zero inplane wavevector component. OTS is observed at the interface between a photonic crystal and a cGST layer (as shown in Figure 3b). We present the numerical calculation of absorption spectra upon normal incidence of light with different *h*_1_ and *h*_2_ in Figure 5. Both structures consisted of four pairs of Ge and ZnS layers. Spectral position of this resonance was highly sensitive to *h*_1_ and *h*_2_. Red-shifting of the absorption peak with increasing *h*_1_ and *h*_2_ was observed. In designing the perfect absorber with this structure, a convenient approach to tune the resonant wavelength is provided. It was also found that total absorption can be maintained over 90%, which provides the opportunity to tune resonance over a wider range of structural parameters. Dependence of the absorption spectrum on *N* was investigated and shown in Figure 6. The light absorption increased, and then kept the maximal value (~99.6%) when *N* was larger than 9.

Figure 7 presents a color-map of absorbance as a function of wavelength and angle of incidence, which is varied from 0° to 90°. Figure 7a,b shows the absorbance in the amorphous phase and crystalline phase for TM polarization while Figure 7c,d gives the absorbance in the amorphous phase and crystalline phase for TE polarization. For both cases, the absorbance was reduced with the increases of incidence angle. The absorbance peak was still larger than 80% at 60°. Moreover, there was a good spectral overlap between the absorbance for transverse electric (TE) and TM polarizations in each phase state. Therefore, the proposed MMA is polarization independent over a wide range of incident angles.

## 4. Conclusions

In summary, a wavelength-selective polarization independent MMA in the MIR range was proposed using the optical Tamm state surface mode near the GST interface. Compared with previous emphasis on three-dimensional structures, no sophisticated lithography was required, and the resonant absorption bands were tuned by switching the phase of PCM GST between amorphous and crystalline states. This work will find great potential applications in cost-effective active photonic devices.

## Figures and Tables

**Figure 1 micromachines-10-00374-f001:**
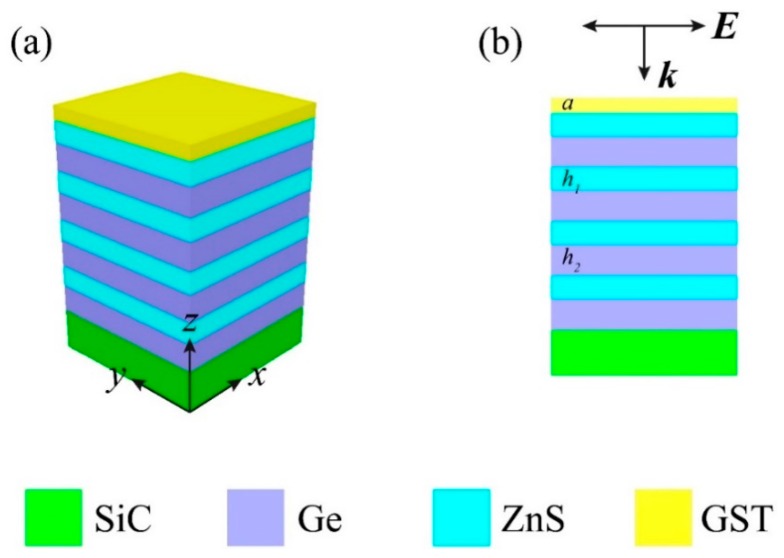
(**a**) Schematic of the initial structure; (**b**) cross-section view of the proposed absorber.

**Figure 2 micromachines-10-00374-f002:**
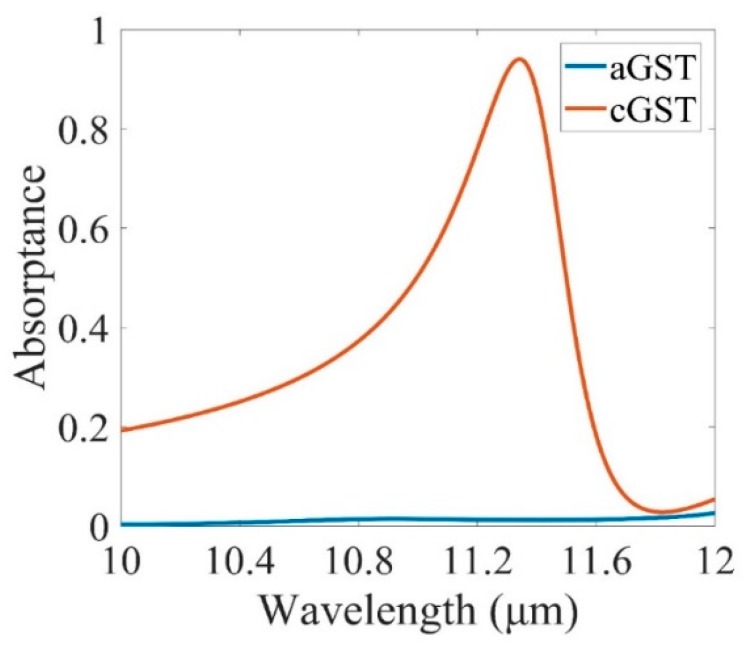
Total absorption at normal incidence for structure with amorphous (aGST) and crystalline (cGST) stable structural phases. The geometrical parameters are originally assumed as *a* = 0.3 μm, *h*_1_ = 0.53 μm, *h*_2_ = 0.81 μm. Unless otherwise stated, they are used throughout the paper.

**Figure 3 micromachines-10-00374-f003:**
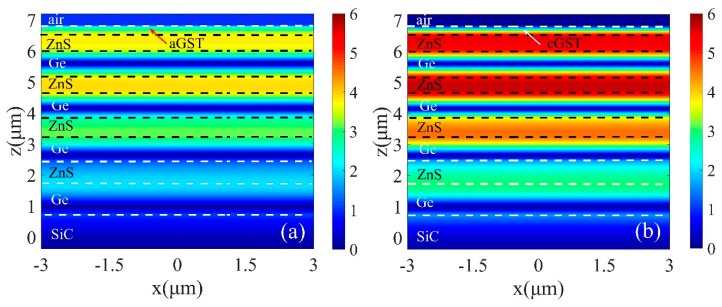
Electric field distribution profiles at the resonance wavelength of 11.34 μm in *x*-*z* plane with (**a**) amorphous (aGST) and (**b**) crystalline (cGST) stable structural phases, respectively.

**Figure 4 micromachines-10-00374-f004:**
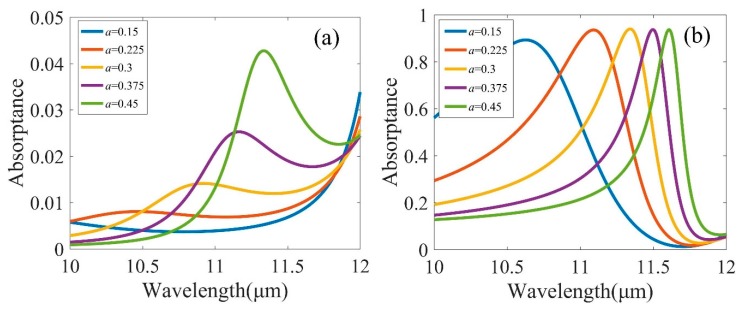
Absorption spectra for various height of (**a**) GST with amorphous (aGST) and (**b**) crystalline (cGST) stable structural phases, respectively. Other geometric parameters were the same as used in Figure 2.

**Figure 5 micromachines-10-00374-f005:**
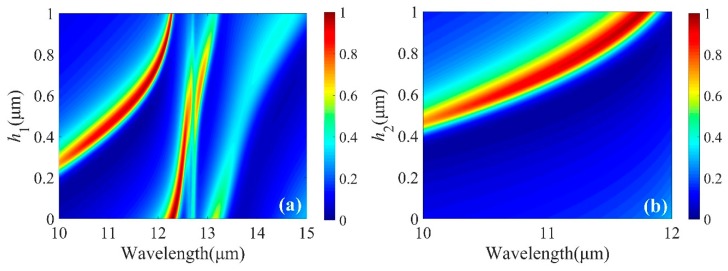
Absorption spectra for various height of ZnS with (**a**) amorphous (aGST) and (**b**) crystalline (cGST) stable structural phases, respectively.

**Figure 6 micromachines-10-00374-f006:**
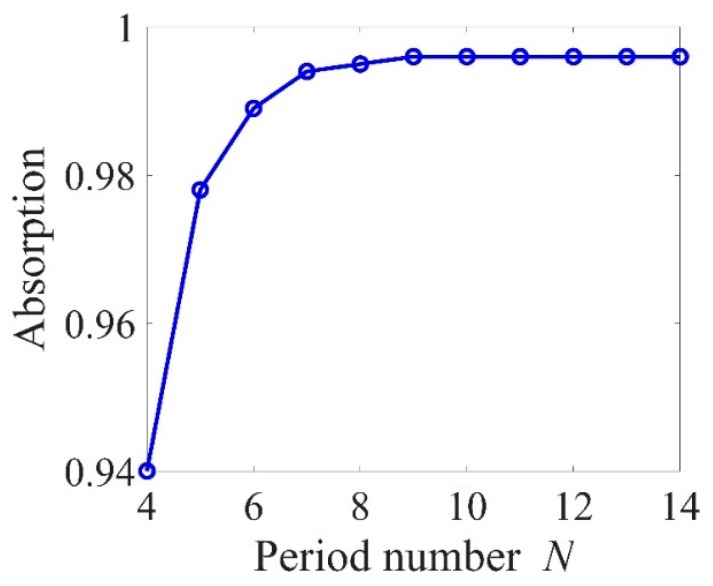
Peak values of light absorption of the structure with period number *N* when *a* = 0.3 μm, *h*_1_ = 0.53 μm and *h*_2_ = 0.81 μm.

**Figure 7 micromachines-10-00374-f007:**
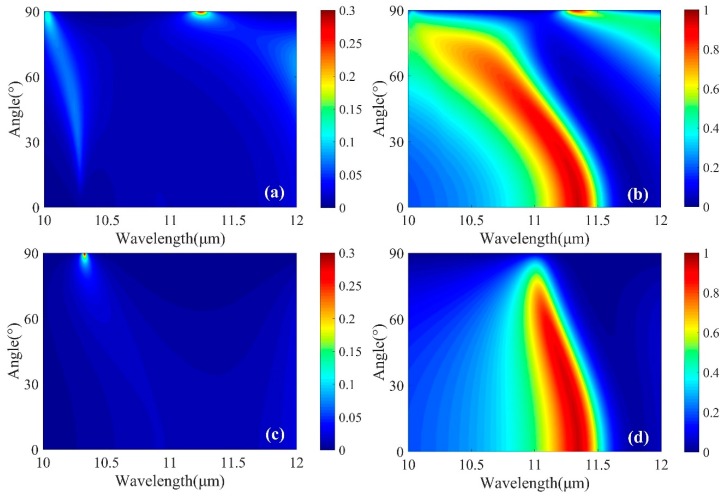
(**a**,**b**) Simulated absorption spectra with different angles of incidence, at the condition of amorphous and crystalline phases for transverse magnetic (TM) polarization, respectively. (**c**,**d**) Simulated absorption spectra with different angles of incidence, at the condition of amorphous and crystalline phases for transverse electric (TE) polarization, respectively. Other geometric parameters are the same as used in Figure 2.

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
