# Peer review of "Intensity Switchable and Wide-Angle Mid-Infrared Perfect Absorber with Lithography-Free Phase-Change Film of Ge2Sb2Te5"

_micromachines, 2019, doi:10.3390/mi10060374_

Round 1
Reviewer 1 Report
In this manuscript, the authors have theoretically investigated a mid-infrared tunable absorber that consists of lithography-free planar multilayered dielectric stacks and germanium antimony tellurium alloy (Ge2Sb2Te5, GST) thin film. Significant tuning of absorption has been achieved by switching the phase of thin layer of GST between amorphous and crystalline states. The novelty is not high enough and the manuscript is not well organized. Thus, the manuscript would need to incorporate the following changes to be fit for publication in Micromachines.
1. In this abstract, the authors claimed “mid-infrared tunable absorber/emitter”. However, the manuscript is mainly focused on absorber. If the designed absorber is used as emitter, the operating temperature should be considered. Please change the statement.
2. In the manuscript, the authors have used constant refractive index to describe GST. However, the real optical constants of cGST and aGST are dispersive. In this way, the authors need to take the dispersive optical constants into consideration and re-calculate the related results.
3. Besides the amorphous and crystalline states, it is also interesting and important to see the absorption spectra when GST is in its intermediate state. Please add the related results and discussions.
4. On Page 3, Line 79-80, “In this work, e ¥ 79 is chosen as 93.75px-1 and g is set as 6.5.” I think this description is wrong. Please double-check it and make the correction.
5. Fig. 5 only has two sub-figures while the caption contains (a-d) four sub-figures. Please correct this issue.
6. Since the manuscript is related to tunable absorbers based on phase-change material, the authors could consider citing the following papers to make it more comprehensive.
Photonics Research, 4(4) 146-152, 2016
Laser Photonics Review, 11, 1700091, 2017
Advanced Optical Materials, 1801709, 2019
Advanced Optical Materials 6 (9), 1701204, 2018
7. The English needs to be improved as there are many errors throughout the manuscript.
Author Response
I have already replied to your review comments one by one. Please refer to the following for details.

Reviewer 2 Report
The researchers showed a numerical simulation and experiments of perfect absorber operating in mid-infrared range using phase-change materials. It has been shown that a multi-layer device absorbs light in the mid-infrared region through simulation and experimental results. Therefore, the paper has adequate technical and scientific results, so is considered suitable for inclusion in this journal. However, minor changes should be made to make better study results.
There are some minor points which could be corrected.
(1) The notation of h1 and h2 is different.
(2) Some units have been omitted or are not well spaced.
Author Response

(The authors gave the same response as above.)

Round 2
Reviewer 1 Report
The authors have addressed all the comments i raised before. The manuscript can be accepted now.